# Association between self-reported alcohol consumption and diastolic dysfunction: a cross-sectional study

Bledar Daka  ,[1] Louise Bennet,[2] Lennart Råstam,[2] Margareta I Hellgren,[3] Ying Li,[4] Martin Magnusson,[5] Ulf Lindblad[2]

[1]Medicine, University of Gothenburg Sahlgrenska Academy, Goteborg, Sweden
[2]Family Medicine, Deaprtment of Clinical Sciences, Lund University, Malmo, Sweden
[3]Medicine, University of Gothenburg, Göteborg, Sweden
[4]Medicine, Sahlgrenska Akademy, Goteborg, Sweden
[5]Lund University, Lund, Sweden

**Correspondence to**
Dr Bledar Daka;
bledar.daka@allmed.gu.se

## ABSTRACT

**Background and objectives**  While alcohol consumption is associated with common risk factors for diastolic dysfunction the independent impact of low levels of alcohol consumption on this condition in a community setting is still unclear.
Thus, the aim of this study was to explore this association in a representative population sample employing optimal echocardiographic techniques.

**Design**  Cross-sectional observational study in community-based population.

**Settings, participants and methods**  Participants between 30 and 75 years of age were consecutively invited to a physical examination, interview, conventional echocardiography, including Tissue Velocity Imaging. Diastolic dysfunction was defined according to the European Society of Cardiology criteria, excluding subjects with ejection fraction <45%, self-reported history of heart failure or atrial fibrillation on ECG. Self-reported alcohol intake using a validated questionnaire was categorised as *no intake, low and medium-high* intake.

**Results**  In total, 500 men and 538 women (mean age 55.4±13) were successfully examined. Diastolic dysfunction was identified in 16% (79/500) of the men and 13% (58/538) of the women. The multivariable adjusted model revealed a strong and independent association between alcohol intake and diastolic dysfunction. In fact, using no alcohol intake as reference, diastolic dysfunction was independently associated with alcohol consumption in a dose-dependent fashion; *low consumption,* OR 2.3 (95% CI 1.3 to 4.0) and *medium-high consumption* OR 3.1 (95% CI 1.6 to 6.2), respectively.

**Conclusion**  There was a significant association between alcohol consumption and diastolic dysfunction starting already at low levels that was supported by a dose-dependent pattern. These results need confirmatory studies and are important in public health policies.

## INTRODUCTION

Based on WHO from 2018 about 57% of the adult population globally are alcohol consumers and 3 million deaths are attributed to alcohol consumption in 2016.[1 2] A recently published meta-analysis could identify better health outcomes in people that had reported low to moderate alcohol consumption.[3] On the other hand, several studies have shown

## STRENGTHS AND LIMITATIONS OF THIS STUDY

⇒ Echocardiography was performed in a large population-based cohort using standardised methods. The re-evaluation of a subsample by a senior cardiologist confirmed the first assessment and the consistency.
⇒ Detailed information on other risk factors including oral glucose tolerance test to define diabetes mellitus was available.
⇒ Validated questionnaires on alcohol consumption was used, however, still self-reported.
⇒ Finally, this was a cross-sectional observational study and no conclusions in causality can be drawn. However, randomised controlled trials in this field are and will remain difficult to perform.

that moderate levels of alcohol (ie, alcohol equalling <1 drink per day (d/d) of alcohol in women and <2 d/d in men) have a protective effect on ischaemic heart disease and chronic heart failure (CHF)[1 4] while heavy episodic drinking negates from these effects.[5] Interestingly, chronic excessive alcohol consumption is also reported to be associated with diastolic dysfunction,[6] which is characterised by disturbed relaxation and increased preload. Diastolic dysfunction represents one out of three criteria for diastolic heart failure[7] and occurs often without impaired physical performance or other symptoms. The condition is common with prevalence in community-based studies varying from a few to 30%[8 9] and is associated with increase in risk to develop heart failure[10]

In patients with hypertension, moderate alcohol consumption have been associated with diastolic dysfunction.[11] Although alcohol consumption is associated with common risk factors for diastolic dysfunction as hypertension[12] and obesity[13] the independent impact of low to levels of alcohol consumption on this condition in a community setting is still unclear. While there is consensus on deleterious effects of heavy drinking, the role of low

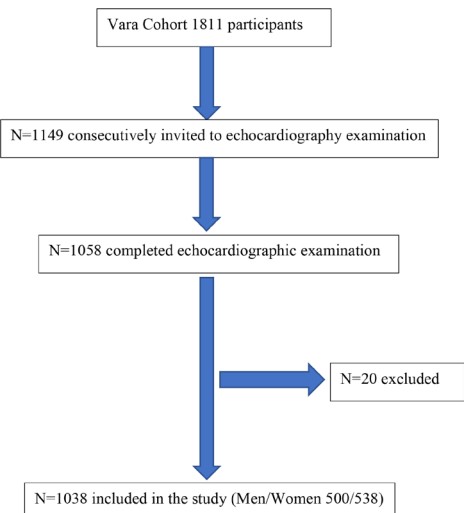

**Figure 1** Study population in the Vara cohort 2002–2003. *Cases excluded due to at least one of the following reasons: arrhythmia/atrial fibrillation n=8, physical characteristics hindering echocardiography n=6, self-reported history of heart failure n=6, ejection fraction<45% n=2 and/or aortic insufficiency n=2.

alcohol consumption on cardiovascular disease (CVD) and CHF risk and diastolic dysfunction is unknown. Also, the mechanisms of a protective effect in low doses and deleterious effect in higher doses have not been explained yet.

Thus, the objective of this study was to explore the associations between alcohol intake and asymptomatic diastolic dysfunction with preserved systolic function in a population-based cohort.

## MATERIALS AND METHODS
### Subjects
This study includes data from a surveillance of the population in Vara 2002–2003, a small community with 16 000 residents in a rural area of south-western Sweden.[14] The sampling, inclusion of participants and echocardiography-examination was conducted as previously described.[15] Briefly, men (500) and women (538) aged 30–75 years were selected from the census register and invited to participate. Individuals below age 50 years were oversampled three times as compared with those who were older. Participation was based on conducting an echocardiography and physical examination, filling in questionnaires and returning blood samples. Subjects with ejection fraction <45%, self-reported history of heart failure, arrhythmia, aortic insufficiency or physical characteristics hindering echo were excluded from the analyses (N=20). The study profile is presented in figure 1.

### Demographic and lifestyle data
Standardised physical examinations were performed and blood samples drawn including oral glucose tolerance tests.[15] Blood pressure was measured in a lying position after 5 min rest. Two measurements were done within

1 min apart and the mean value was used. Diagnosis of hypertension was defined in accordance with international guidelines based on medical history of hypertension or three consecutive blood pressure measurements equal or higher than 140/90 mm Hg.[16] Diabetes was defined based on WHO recommendations.[17] Information on previous CVD was collected through questionnaires. Information on smoking habits was also collected through questionnaires and participants were categorised as never smokers, former smokers and current smokers.

### Alcohol intake
Participants were asked to estimate the number of days they had been drinking alcohol the last month and the average amount of alcohol they had consumed during those days. Based on validated questionnaires[18] for self-reported alcohol intake an estimation on grams per week of alcohol consumption was calculated for each participant. Based on quartiles of this estimated weekly alcohol consumption, the participants were divided into three groups. Non-drinkers were participants that reported no alcohol consumption last month. Highest quartile (>56 g/week) was defined as medium-high consumers and those with alcohol consumption ≤56 g/week were defined as low consumers. One drink of alcohol was estimated equivalent to 15 g of alcohol.[19]

Level of leisure time physical activity (LTPA) was self-reported in four possible levels: *Sedentary LTPA*, that is, mostly physically inactive, *less strenuous LTPA* (walks, cycling, gardening, etc) <4 hours a week (h/w); *medium LTPA*, that is, less strenuous LTPA>4 h/w or *high LTPA*, that is, strenuous LTPA (jogging, swimming, tennis, etc) ≥2 h/w.[20] Due to the reduced number of individuals in the strenuous category we grouped groups 3 and 4 together and defined it as reference.

### Echocardiographic methods
All participants were examined by the same senior cardiologist, using echocardiography (General Electrics VingMed S 5 System, operating with a 3.5 MHz-probe).[15] The same cardiologist performed all the measurements and had no access to information on alcohol drinking habits. Data was stored in the EchoPAC system for playback analysis and measurement. Measurements for left ventricular (LV) calculations were taken from the Guidelines of the American Society of Echocardiography.[21] A two-dimensional echocardiogram was formed from the left parasternal and apical windows for measurements. The ejection fraction (EF) was calculated from apical four and two chamber views with longitudinal systolic shortening mean index/atrioventricular plane displacement technique in addition to a semi-quantitative visual estimate method.[22 23] The ventricular diastolic function was based on several parameters and their respective normal ranges were defined for different age categories.[24] The E-wave-peak (early filling) to A-wave-peak (atrial filling), and the ratio were then determined. The isovolumetric relaxation time, from the closure of the aortic valve to

the opening of the mitral valve, was also measured. To indirectly measure distensibility in the LV, the deceleration time, the peak E to baseline-slope, was measured. Tissue Velocity Imaging was used in both the septum and in the more stable lateral wall to detect pseudonormalisation. The patient was also told to perform the Valsalva manoeuvre during registration.

Diastolic dysfunction was defined according to the recommendations of the European Society of Cardiology[24 25] when evidence of abnormal LV diastolic relaxation, filling, diastolic distensibility and diastolic stiffness were found in the presence of normal or mildly reduced LV systolic function (EF≥45%). Diastolic dysfunction was categorised into three levels as: *impaired relaxation*, that is, early diastolic dysfunction; *pseudonormal*, that is, a more severe condition indicating increasing LV filling pressures or *restrictive filling pattern*. Diastolic dysfunction was then further dichotomised as being normal or not with impaired relaxation, pseudonormal or restrictive filling patterns categorised as diastolic dysfunction. Another senior cardiologist—not otherwise involved in the project and unaware of the findings in the first examination—reviewed the diagnostic criteria and the echocardiographic methods and re-examined a random sample (n=30) of the study sample using stored images. The results were consistent showing identical categorisation.

Left ventricular mass was defined as: LVM=0.8 × (1.04 ((LV internal dimension+posterior wall thickness+septal thickness)$^3$ – (LV internal dimension in diastole)$^3$)) + 0.6 g. LVM was then indexed for height$^{2.7}$ (LVM-index), and left ventricular hypertrophy (LVH) was defined as LVM-index ≥47.3 g/m$^{2.7}$ based on guidelines.[26 27]

## Statistical analysis

Standard methods were used for descriptive statistics which were presented in terms of mean and SD for all continuous variables and counts with percentage of categorical variables. All the statistical analysis were performed using program R V.4.1.2. Possible confounding was addressed by stratification and multivariate analyses. Theoretical multiple logistic regressions were used to evaluate the association between diastolic dysfunction and alcohol consumption categories, with adjustments for different sets of covariates. In the theoretical models we included traditional confounding factors as age and gender. Current smoking status and LTPA are lifestyle factors with strong covariation with alcohol consumption and that can influence the risk for diastolic dysfunction. Traditional determinants of diastolic dysfunction such as hypertension, type 2 diabetes and obesity were also included in the model. Model 1 includes only age and gender as traditional confounding factors. Model 2 adds smoking status and LTPA as lifestyle factors, which could be regarded as possible confounders or risk factors for outcome. Finally, Model 3 includes hypertension, type 2 diabetes and body mass index (BMI), which could be considered as mediators or strong determinants of diastolic dysfunction.

By presenting all three models the reader will have the possibility to judge the extent to which a change of the effect measure reflects removing a confounding effect and the extent to which the change of the effect measure reflects the degree of possible mediation.

These associations were expressed as ORs with 95% CIs. All tests were two-sided and statistical significance was assumed at p<0.05.

### Ethical considerations

The investigation conforms to the principles outlined in the Declaration of Helsinki. The participants provided written informed consent to their participation in the health survey, and a further written informed consent was obtained before undergoing an ultrasound-Doppler examination of the heart.

### Patient and public involvement

A meeting in the study centre and the primary healthcare centre was organised prior to the start of the study. A discussion with caregivers was organised to meet their concerns and ideas for the study but also to organise the care for individuals with pathological findings in the study. An open house event for the public was organised to inform and receive questions and suggestions. The results of the study are disseminated within the healthcare centres and health professionals.

### Data availability statement

The data generated in the current study are available on reasonable request, considering the sensitivity and confidentiality of the data. Due to the nature of the data, which contains personally identifiable information and sensitive information, it cannot be openly shared in a public repository.

## RESULTS

The mean age of the population was 55.4±13 years. In total, 79 of 500 men (16.5%) and 58 of 538 (12.9%) women were diagnosed with diastolic dysfunction (p=0.037). A total of 94% of the cases were categorised as *impaired relaxation* (74 men, 55 women) and 6% as *pseudonormal* (5 men, 3 women). There were no cases of severe *restrictive* filling patterns.

Alcohol consumption was common in both genders; men 84.5% (412/486) and women 68.8% (375/527), however, significant differences were observed between men and women with women being non-drinkers to a larger extent. In this observational study, non-drinkers were older, had higher blood pressure and had more often diabetes, however. Being allocated in one of groups based on alcohol consumption was not associated with diastolic dysfunction (p=0.500) (table 1). Meanwhile, among subjects not using alcohol, a higher rate of LVH was observed. Characteristics of the study population as well as echocardiographic characteristics related to level of alcohol intake are presented in table 1.

**Table 1** Characteristics of participants related to level of alcohol consumption

| Variables | Alcohol consumption | | | P value* |
| --- | --- | --- | --- | --- |
| | Non-consumers n=226 (22%) | Low n=531 (53%) | Medium-high n=254 (25%) | |
| Diastolic dysfunction† | 35 (15.5) | 66 (12.4) | 32 (12.6) | 0.500 |
| Age (years)‡ | 55.4 (12.8) | 49.5 (11.5) | 48 (10.7) | 0.050 |
| Women† | 152 (67.3) | 289 (54.4) | 85 (33.5) | <0.001 |
| BMI (kg/m$^2$)‡ | 27.8 (5.4) | 26.8 (4.2) | 27 (3.9) | <0.001 |
| SBP (mm Hg)‡ | 128.4 (18.9) | 123.6 (17.1) | 123.2 (15.6) | 0.001 |
| DBP (mm Hg)‡ | 71.1 (9.7) | 71.1 (10.3) | 72.1 (10.1) | 0.300 |
| HDL (mmol/L)‡ | 1.2 (0.3) | 1.3 (0.3) | 1.3 (0.3) | 0.100 |
| TG (g/L)‡ | 1.4 (0.7) | 1.3 (0.8) | 1.3 (0.7) | 0.700 |
| Level of physical activities | | | | 0.020 |
| Mobile | 57 (25.2%) | 191 (36.0%) | 90 (35.4%) | |
| Medium | 147 (65%) | 312 (58.8%) | 149 (58.7%) | |
| Sedentary | 22 (9.7%) | 28 (5.3%) | 12 (5.9%) | |
| Hypertension† | 61 (27.0) | 93 (17.5) | 33 (13) | <0.001 |
| Diabetes† | 29 (12.8) | 34 (6.4) | 10 (3.9) | <0.001 |
| Smoking habits† | 39 (17.3) | 92 (17.3) | 54 (21.4) | 0.300 |
| E/A ratio | 1.2 (0.4) | 1.4 (0.4) | 1.4 (0.4) | <0.001 |
| IVRT | 98.4 (21.9) | 91.6 (20.1) | 93.2 (21.2) | <0.001 |
| Ejection fraction | 72.2 (9.6) | 74.4 (8.4) | 74.4 (7.8) | 0.005 |
| Septal wall (cm) | 1 (0.2) | 0.9 (0.2) | 0.9 (0.2) | 0.008 |
| Posterior wall (cm) | 1 (0.3) | 0.9 (0.2) | 1 (0.5) | 0.010 |
| LAD (cm) | 3.3 (0.5) | 3.2 (0.5) | 3.3 (0.5) | 0.040 |
| LV-mass (median and 5th, 95th percentiles) | 37.8 (23.4, 62) | 34.5 (23.8, 55.6) | 35.1 (23.9, 57.5) | 0.050* |
| LVH | 52 (23.0%) | 70 (13.2%) | 32 (12.6%) | 0.001 |

*P values are based analysis of variance if the variables are continuous or based on $\chi^2$ test if the variables are binary. Medium-high is defined as consumption >56 g/week and low those with alcohol consumption ≤56 g/week.
†Data is presented as counts (per cent).
‡Data are presented with mean (SD).
§
BMI, body mass index; DBP, diastolic blood pressure; E/A ratio, The ratio of peak velocity blood flow from left ventricular relaxation in early diastole (the E wave) to peak velocity flow in late diastole caused by atrial contraction (the A wave); HDL, high density lipoprotein-cholesterol; IVRT, isovolumic relaxation time; LAD, left atrial diameter; LV, left ventricular; LVH, left ventricular hypertrophy; SBP, systolic blood pressure; TG, triglycerides.

In a multivariable theoretical model, adjusted for possible confounders and mediators in the association between alcohol consumption and diastolic dysfunction, we found that alcohol consumption was significantly associated with increased risk of diastolic dysfunction. Interestingly, even low alcohol consumption was strongly associated with increased risk for diastolic dysfunction (OR=2.2 95% CI 1.3 to 3.9). The estimates were even higher for participants with moderate to high alcohol consumption (OR=3.1 95% CI 1.6 to 6.2) with a strong significant trend (p=0.001). Similar results were observed when adjustments for systolic blood pressure were included in the final models (low consumption: OR=2.3 95% CI 1.3 to 4.1; moderate to high alcohol consumption OR=3.1 95% CI 1.6 to 6.2). In another model we tested also to adjust in the final model for high density lipoprotein-cholesterol on the top of adjustments for age, gender, hypertension, type 2 diabetes, BMI, smoking habits and leisure time physical activities and the association was still strong and significant (low consumption: OR=2.4 95% CI 1.4 to 4.4; moderate to high alcohol consumption OR=3.4 95% CI 1.7 to 6.9). Exploring in the same models the association between alcohol consumption and left atrial diameter no significant differences were observed between group with no consumption and groups with low and moderate to high (age and sex adjustments: low consumption: mean difference (cm)=−0.04 95% CI −0.11 to 0.02; moderate to high alcohol consumption mean difference (cm)=−0.02 95% CI −0.10 to 0.07; fully adjusted model: low consumption: mean difference (cm)=−0.02 95% CI −0.08 to 0.04; moderate to high alcohol consumption mean difference (cm)=0.0 95% CI −0.07 to 0.08). There were no

**Table 2** Association between risk factors and diastolic dysfunction

| | OR (95% CI) |
|---|---|
| **Model 1** | |
| Age | 1.16 (1.13 to 1.19) |
| Men vs women | 1.34 (0.87 to 2.07) |
| Alcohol consumption | |
| None | 1 (ref) |
| Low vs none | 1.82 (1.08 to 3.12) |
| High vs none | 2.39 (1.26 to 4.58) |
| **Model 2** | |
| Age | 1.16 (1.13 to 1.19) |
| Men vs women | 1.46 (0.94 to 2.27) |
| Current smoking | 1.39 (0.78 to 2.42) |
| Physical activity | |
| Active | 1 (ref) |
| Medium vs active | 1.6 (0.97 to 2.69) |
| Sedentary vs active | 4.4 (1.93 to 9.96) |
| Alcohol consumption | 1 (ref) |
| Low vs non | 2.03 (1.19 to 3.5) |
| High vs non | 2.68 (1.4 to 5.19) |
| **Model 3** | |
| Age | 1.14 (1.11 to 1.18) |
| Men vs women | 1.63 (1.02 to 2.63) |
| BMI | 1.07 (1.01 to 1.11) |
| Hypertension | 2.68 (1.65 to 4.33) |
| Type 2 diabetes | 1.88 (1.00 to 3.53) |
| Current smoking | 1.76 (0.96 to 3.15) |
| Leisure time physical activity | |
| Medium vs active | 1.28 (0.75 to 2.23) |
| Sedentary vs active | 2.82 (1.15 to 6.84) |
| Alcohol consumption | |
| None | 1 (ref) |
| Low vs none | 2.25 (1.29 to 4.02) |
| High vs none | 3.06 (1.55 to 6.15) |

Multivariable logistic regression analyses are used to investigate the association between explanatory variables and diastolic dysfunction. In model 1 adjustments for age and gender, in model 2 adjustments for age, gender, current smoking, leisure time physical activities, in model 3 adjustments for age, gender, BMI, hypertension, type 2 diabetes, current smoking, leisure time physical activities are computed. Women are the reference group in the analyses men versus women and physically active group is the reference in the analyses of leisure time physical activity.
BMI, body mass index.

**Table 3** Association between risk factors and left ventricular hypertrophy

| | OR (95% CI) |
|---|---|
| Age | 1.08 (1.06 to 1.1) |
| Men vs women | 1.16 (0.75 to 1.78) |
| BMI | 1.16 (1.11 to 1.22) |
| Hypertension | 3.20 (2.05 to 5.01) |
| Type 2 diabetes | 1.30 (0.70 to 2.38) |
| Current smoking | 1.53 (0.89 to 2.57) |
| Leisure time physical activity | |
| Medium vs mobile | 0.96 (0.60 to 1.56) |
| Sedentary vs mobile | 1.18 (0.50 to 2.69) |
| Alcohol consumption | |
| Non-drinkers | 1 (ref) |
| Low vs non | 1.08 (0.59 to 1.98) |
| High vs non | 1.53 (0.89 to 2.57) |

Multivariable logistic regression analyses are used to investigate the association between explanatory variables and left ventricular hypertrophy.
BMI, body mass index.

Similar analyses were computed using LVH as outcome, but no significant association was found with alcohol consumption (table 3).

## DISCUSSION

In this cross-sectional observational study, we found a strong and independent association between self-reported alcohol consumption and diastolic dysfunction. Interestingly, our data show an increased risk of having diastolic dysfunction already in subjects with low level of alcohol consumption. Moreover, in gender specific analyses no gender differences were observed suggesting a similar effect of alcohol in both men and women. Despite these findings, no significant association between alcohol intake and LVH were observed. Like previous observations, age, gender, hypertension, leisure time physical activity levels and BMI were all strongly and independently associated with diastolic dysfunction.

These findings are not in line with most of the observations on the role of alcohol in heart failure. In fact, population-based studies have shown that *moderate* levels of alcohol consumption (ie, <1 d/d in women and <2 d/d in men) have a protective effect on ischaemic heart disease and decreases the incidence of CHF. [1 3 28] The beneficial effects of a low to moderate amount of alcoholic intake are not reported by all epidemiological studies.[29 30] A 'healthy cohort effect' is reported as one plausible explanation to the favourable effects of moderate consumption[29 30] where the confounding effects of, for example, physical activity, mental health, psychosocial, socioeconomic and social network-related factors seldom are accounted for.[30] Moreover, in a recent observation in HUNT study[31] no

significant differences in these associations were observed between men and women (p for interaction >0.25). High age, male sex, hypertension, high BMI and a low level of leisure time physical activity were all also independently associated with diastolic dysfunction (table 2).

protective effect of low-alcohol consumption on LV function was observed, however, a positive linear association was observed between alcohol consumption and LVM and end diastolic diameter. The findings of HUNT-study are in line with our findings however the significance in the association was lost in their multivariable analyses. Interestingly, before the adjustments in our cohort higher prevalence of diastolic dysfunction was observed in non-drinkers. These differences were not significant. The same group had also higher age, lower level of physical activities, higher systolic blood pressure and higher prevalence of hypertension and diabetes. It was first after the adjustments for these factors that a positive linear association between alcohol consumption and diastolic dysfunction was observed. These findings suggest no protective effect of alcohol at low levels. Also, alcohol seems not to be related with better diastolic dysfunction, but healthy and younger subjects are more prone to drink alcohol and they have of course better diastolic function but not of alcohol consumption.

Several alcohol metabolites are identified as particularly toxic to the myocardium: ethanol, acetaldehyde (the metabolic product of alcohol) and fatty acid ethyl esters.[32 33] Alcohol also impairs the transport of calcium ions from the sarcoplasma to the sarcoplasmatic reticulum[34 35] resulting in diastolic dysfunction characterised by delayed myocardial relaxation[36] and impaired LV diastolic filling. Moderate versus high doses of alcohol affect myocardial function differently through oxidative stress, haemostatic factors, levels of C-reactive protein, interleukins and tumour necrosis factor alpha.[29]

There have been observations that suggest that patterns of drinking and food intake might explain at least partially the damages of alcohol in myocardium.[5 37 38] In fact, in the Mediterranean countries with a low prevalence of CVD, alcohol is commonly consumed with meals as compared with Russia with a high prevalence of CVD where binge drinking is much more frequent.[30] However, some studies have shown there is evidence that alcoholic cardiomyopathy can be developed even in the absence of malnutrition[39] while these deleterious effects are exaggerated.[40 41]

Interestingly, our results did not show any significant association with LVH. One reason for the lack of association might be the relatively low number of individuals with high consumption in this cohort. Measurable structural remodelling might need a higher level of alcohol consumption. In fact, a large population study in South Korea with a significantly larger number of participants compared with our study, could find a strong association between alcohol consumption and LVH.[42]

### Strengths and limitations
Echocardiography was used in this large population-based cohort using standardised age-dependent criteria to identify diastolic dysfunction based on recommendations by international expert groups. The re-evaluation of a subsample by a senior cardiologist confirmed the first assessment and the consistency in assessment of prevalence of diastolic dysfunction and in identified risk factors between our findings and other comprehensive surveys using Doppler techniques, lend further support to the validity of our data.

The characteristics of alcohol consumers and prevalence of alcohol consumption was in accordance with other comparable community-based surveys.[43 44] This, however, might be a source of underestimation of the alcohol consumption. In fact, in several studies there is an underestimation of self-reported alcohol consumption[45] and there is a need for more objective and reliable measures of alcohol consumption. However, the underestimation should be more frequent and mostly in the group with higher consumption. This would not have an impact on our results in the association found between low alcohol consumption and diastolic dysfunction. Moreover, the findings are based on a rural population in Sweden and differences in these associations might appear when other populations are considered, however the consistency of the results with previous findings indicate that these findings might apply in other populations. Finally, this was a cross-sectional observation and no conclusions in causality can be drawn and the residual confounding is not possible to rule out.

### CONCLUSION
Our study revealed a significant association between alcohol consumption and diastolic dysfunction starting already at low levels that was supported by a dose-dependent pattern. Additional prospective surveys with adequate measurement of alcohol consumption and possible confounders are needed to improve our understanding of the underlying relationships and mechanisms. Our results might be of importance in the development of public health policies and preventive strategies.

**Acknowledgements** We are indebted to all participants in the Vara Cohort. Without them this study would have not been possible. The Knut and Alice Wallenberg foundation is recognised for generous support.

**Contributors** All authors read and approved the final manuscript. Research idea and study design: LB, BD, UL, MH. Data acquisition: UL, LR, BD, MH. Data analysis and interpretation: LB, UL, BD, MM, MH. Statistical analyses: LB, BD, YL, UL. BD is responsible for the overall content. Each author contributed important intellectual content during manuscript drafting or revision, accepts personal accountability for the author's own contributions and agrees to ensure that questions pertaining to the accuracy or integrity of any portion of the work are appropriately investigated and resolved.

**Funding** This work was supported by: The Swedish Science Council; Sahlgrenska Academy, University of Gothenburg; Skaraborg Institute; The Health & Medical Care Committee of the Regional Executive Board of the Region Västragötalands region VGFOUREG-86294, VGFOUREG-969459, ALFGBG-966255, ALFGBG-971147; Skaraborg Primary Care; Skane Region; Lund University; University Hospital of Malmoe donation Foundation; Påhlsson Foundation; Craford Foundation; Lundberg Foundation, the Swedish Heart and Lung foundation and the Wallenberg Center for Molecular Medicine, Lund University; and Lund University. The funding source was not involved in the work with the article.

**Competing interests** None declared.

**Patient and public involvement** Patients and/or the public were not involved in the design, or conduct, or reporting, or dissemination plans of this research.

**Patient consent for publication** Not applicable.

**Ethics approval** This study involves human participants and was approved by Research ethics committee at the University of Gothenburg DNR Ö199-01. Participants gave informed consent to participate in the study before taking part.

**Provenance and peer review** Not commissioned; externally peer reviewed.

**Data availability statement** Data are available upon reasonable request. Data are available upon reasonable request as it contains pseudonimised information.

**ORCID iD**
Bledar Daka http://orcid.org/0000-0002-7722-1528

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
