## [Reviewer comments · BMJ Open]

ARTICLE DETAILS

TITLE (PROVISIONAL)	Association between self-reported alcohol consumption and diastolic dysfunction, a cross-sectional study
AUTHORS	Daka, Bledar; Bennet, Louise; Råstam, Lennart; Hellgren, Margareta; Li, Ying; Magnusson, Martin; Lindblad, Ulf

VERSION 1 – REVIEW

REVIEWER	Wong, Jorge McMaster University
REVIEW RETURNED	16-Dec-2022

GENERAL COMMENTS	Dr. Daka and colleagues submit their manuscript entitled “Association between self-reported alcohol consumption and diastolic dysfunction, a cross-sectional study” for peer review. The patient population consists of 1038 individuals from a rural Swedish community. In a cross-sectional analysis, the association between categorical alcohol consumption and diastolic dysfunction was examined. Diastolic dysfunction was determined from echocardiographic data and alcohol consumption was self-reported. The authors found that compared to those with “no alcohol” consumption, low and medium-high alcohol was associated with an increased odds of diastolic dysfunction. Increasing alcohol consumption was not associated with increased odds of LVH. The quality of the writing is fair with multiple errors in punctuation and grammar. A significant weakness is the lack of novelty, as the relationship between alcohol and diastolic dysfunction has been described in individuals consuming low levels of alcohol in much larger studies before. My comments are as follows: Abstract: 1. The authors claim that the “independent impact of low to medium-high levels of alcohol consumption on this condition in a community setting is still unclear.” Low to medium-high refers to a very broad amount of alcohol consumption and I suggest that they drop the term.2. The mean age is listed in the “Setting” section of the abstract. This should be moved to the results.3. A p-value is presented when the prevalence of diastolic dysfunction is compared between men and women. This is not the focus of the paper and I suggest to delete it.4. The authors do not comment on the lack of association with LVH in their abstract, and this should be mentioned.
---

	5. The final statement is too strong. There are reported benefits to low levels of alcohol that need to be balanced as well. Methods: 6. Calling the control group “non-drinkers” is misleading since some individuals in this group consumed alcohol. Why did the authors elect to included drinkers in this group? 7. Ascertainment of the outcome: were echocardiographers blinded to patients alcohol consumption. If not, bias can be introduced. 8. Confounders: How was hypertension adjusted for in multivariable models? Hypertension may have been underdiagnosed. Models should be additionally adjusted for baseline systolic blood pressure. HDL cholesterol has been linked with increased diastolic dysfunction and should be added to multivariable models. BMI was also adjusted for in models correct? This is not clearly stated in the Methods. Results/Discussion 9. LA enlargement is a hallmark of diastolic dysfunction. Was LA volume measured. Was alcohol associated with increased LA volume as well? 10. Page 9: causality cannot be shown in a cross-sectional study, yet the authors state that there is “a possible reversal causality,” presumably between diastolic dysfunction and alcohol. This statement is not clear. 11. Is there any data on types of alcohol? Is the effect on diastolic dysfunction different based on the type of alcohol consumed? 12. This reference should be added: Catena et al. Moderate Alcohol Consumption Is Associated With Left Ventricular Diastolic Dysfunction in Nonalcoholic Hypertensive Patients. https://doi.org/10.1161/HYPERTENSIONAHA.116.08145 13. A significant weakness is that the study is cross-sectional and the findings may be due to other unmeasured factors that are not adjusted for. This should be added as a limitation. 14. The population is from rural Sweden and the results may not thus be generalizable to other populations. 15. A broader discussion balancing the benefits and harms of alcohol consumption is needed. Minor: There are several grammatical errors and errors in punctuation, which require correction. Abbreviations such as WHO, UCG, need to be explained in full in brackets after the terms.
--	---

REVIEWER	Cesaroni, Giulia ASL Roma 1, Dept. of Epidemiology- Regional Health Service
REVIEW RETURNED	09-Feb-2023

GENERAL COMMENTS	The paper “Association between self-reported alcohol consumption and diastolic dysfunction, a cross-sectional study” is a population-based study on alcohol as a risk factor for diastolic dysfunction. Although the data are from a surveillance of the population in 2002-03, given the high prevalence of both diastolic dysfunction and alcohol consumption, accumulating evidence of an association is important for future prevention. Here follow some points that, if addressed, could improve the paper. Major revisions My major concern about the paper is the statistical approach.
--

	1) The Authors present only the multivariable analyses. It would be useful to see in the same table of multivariable results the results of the models adjusted for age and sex only. 2) The hypothesis of the role of each variable should be stated and the analysis should be performed accordingly. Which are the confounders? Which the possible effect modifiers? And which are the mediators of the association between alcohol and diastolic dysfunction? 3) If you think that hypertension, diabetes and BMI are possible mediators, you should perform a mediation analysis. 4) If you think that sex is a possible effect modifier, you should test it and, in any case, you should show the results from stratified analyses (if the interaction is not statistically significant you can show the results in the online material). Finally, in the limitation section the authors should address the fact that the dataset is quite dated, and that the information on alcohol consumption is cross-sectional, not related to the past habit (it could be useful to have in the paper the specific question). Minor revisions Abstract. Add p-trend<0.001 in line 39. Since the confidence intervals overlap it is difficult to believe that there is a dose-response relationship without a p-trend. Page 3. Spell out OGTT Page 5. Line 12. Spell out UCG Page 6. Lines 53-54. Specify in the text the year of the ESC guidelines, and why you have chosen that edition of guidelines. Page 7. Lines 21-27. Reference 25 is not appropriate (there is not a formula, but other references), and the formula is not correctly written in a mathematical notation. For example, could 2.7 be 2.7 ? Please rewrite it and better explain the background of LVM. Page 7. Lines 31-45. Page 8. Line 20. It is not clear what the three groups are. Page 8. Lines 47-49. Please give the p-trend for alcohol. Page 9. Line 11. If you comment a gender specific result, you should give the reader the table too. Page 20. Table 1. I suggest to specify all the categories of categorical variables i.e., women/men, hypertension yes/no, etc. Moreover, please specify the measurement unit from E/A ratio to LVM. I imagine that LVH is LVHa
--	---

VERSION 1 – AUTHOR RESPONSE

Reviewer: 1

Dr. Jorge Wong, McMaster University

Comments to the Author:

1. A significant weakness is the lack of novelty, as the relationship between alcohol and diastolic dysfunction has been described in individuals consuming low levels of alcohol in much larger studies before.

Answer: Previous studies in different cohorts have reported different results with some studies reporting positive effects of low-alcohol consumption thus observational studies are important for better understanding of a possible role of alcohol in CVD. This is also explained in introduction second paragraph: “Although alcohol consumption is associated with common risk factors for diastolic dysfunction as hypertension(11) and obesity(12) the independent impact of low to levels of alcohol

consumption on this condition in a community setting is still unclear. While there is consensus on deleterious effects of heavy drinking the role of low alcohol consumption on CVD and CHF risk and diastolic dysfunction is unknown”

My comments are as follows:

Abstract:

1. The authors claim that the “independent impact of low to medium-high levels of alcohol consumption on this condition in a community setting is still unclear.” Low to medium-high refers to a very broad amount of alcohol consumption and I suggest that they drop the term.

Answer: We agree, the word medium indicate inclusion of half the distribution which is not the case and not our intention. In consequence we renamed those with the highest consumption as moderate to high consumers, which we think will help the reader to understand. To further emphasize this, we added percentages of the study population for the categories of alcohol consumption in the head of table 1.

2. The mean age is listed in the “Setting” section of the abstract. This should be moved to the results. Done.

3. A p-value is presented when the prevalence of diastolic dysfunction is compared between men and women. This is not the focus of the paper and I suggest to delete it.

Agree, is changed now.

4. The authors do not comment on the lack of association with LVH in their abstract, and this should be mentioned.

Answer: Although this is an important point, the main aim with this study was to investigate the association between alcohol consumption and diastolic dysfunction therefore we chose to take this discussion in the main body of the manuscript before strength and limitation paragraph but difficult to find place in the abstract due to the word limits.

5. The final statement is too strong. There are reported benefits to low levels of alcohol that need to be balanced as well.

Answer: We have changed that into a more balanced statement, and it is now: “These results need confirmatory studies and are important in public health policies”

Methods:

6. Calling the control group “non-drinkers” is misleading since some individuals in this group consumed alcohol. Why did the authors elect to include drinkers in this group?

Answer: The questionnaire used in this study used questions on the alcohol consumption of the last month to estimate the level amount of alcohol in gram per week. Based on that we characterized individuals as non-drinkers when they did not consume any alcohol at all during the last month however, some of the participants might have consumed alcohol exceptionally the last 12 months. We have changed this in the manuscript and it is “Non-drinkers were participants that reported no alcohol consumption the last month.”

7. Ascertainment of the outcome: Were echocardiographers blinded to patients’ alcohol consumption. If not, bias can be introduced.

Answer: The same echocardiographer performed all measurements. The person was not involved in the collection of the data on lifestyle and did not have access to that. We have added in the methods “The same cardiologist performed all the measurements and had no access on information on alcohol drinking habits when evaluating echocardiographies.

8. Confounders: How was hypertension adjusted for in multivariable models? Hypertension may have been underdiagnosed. Models should be additionally adjusted for baseline systolic blood pressure.

Answer: The study has defined hypertension in accordance with current guidelines and the methods for the measurements of blood pressure have been clarified in the methods. We completed the analyses with one model including systolic blood pressure measurement and this is included in the results and it is now: "Similar results were observed when adjustments for systolic blood pressure were included in the final models (Low consumption: OR= 2.3 95% CI 1.3-4.1; moderate to high alcohol consumption OR= 3.1 95% CI 1.6-6.2)"

9. HDL cholesterol has been linked with increased diastolic dysfunction and should be added to multivariable models.

Answer: We completed the analyses with one model including HDL cholesterol measurement and this is included in the results and it is now: "In another model we tested also to adjust in the final model for HDL-cholesterol on the top of adjustments for age, gender, hypertension, type 2 diabetes, BMI, smoking, hypertension, diabetes and leisure time physical activities and the association was still strong and significant (Low consumption: OR= 2.4 95% CI 1.4-4.4; moderate to high alcohol consumption OR= 3.4 95% CI 1.7-6.9)."

10. BMI was also adjusted for in models correct? This is not clearly stated in the Methods.

Answer: BMI is part of the model. Thank you. Done.

Results/Discussion

11. LA enlargement is a hallmark of diastolic dysfunction. Was LA volume measured. Was alcohol associated with increased LA volume as well?

Answer: We agree in the importance of information on LA enlargement however this was not available for us in this study as there was no information on LA-area nor LA-volume. However, we have data on left atrial diameter and there were no significant differences in the left atrial diameter were observed when we compared groups with alcohol consumption. In the manuscript we added in the result section: "Exploring in the same models the association between alcohol consumption and left atrial diameter no significant differences were observed (data not shown)."

12. Page 9: causality cannot be shown in a cross-sectional study, yet the authors state that there is "a possible reversal causality," presumably between diastolic dysfunction and alcohol. This statement is not clear.

Answer: Agree. We have deleted the part of the sentence on reverse causality.

13. Is there any data on types of alcohol? Is the effect on diastolic dysfunction different based on the type of alcohol consumed?

Answer: The main question in this manuscript was to investigate the association between total amount of alcohol consumption and diastolic dysfunction, with no specific regards on the type of beverages. In the specific analyses for type of alcohol, there were slight differences in the strength of the association as these associations were stronger for liquor and slightly weaker for beer and wine. Such fragmentation however increased the risk for type II error thus we chose to analyze for all alcohol intake. A larger cohort would be more suitable to answer the question about differences in the association between different alcohol types.

14. This reference should be added: Catena et al. Moderate Alcohol Consumption Is Associated With Left Ventricular Diastolic Dysfunction in Nonalcoholic Hypertensive Patients.

<https://doi.org/10.1161/HYPERTENSIONAHA.116.08145>

Answer-This reference is added in the introduction "In patients with hypertension moderate alcohol consumption have been associated with diastolic dysfunction (11)."

15. A significant weakness is that the study is cross-sectional and the findings may be due to other unmeasured factors that are not adjusted for. This should be added as a limitation.

Answer: Agree. In the section strengths and limitations we have added " Finally, this was a cross sectional observation and no conclusions in causality can be drawn and the residual confounding is not possible to rule out."

16. The population is from rural Sweden and the results may not thus be generalizable to other populations.

Answer: Agree. We have added "Moreover, the findings are based on a rural population in Sweden and differences in these association might appear when other population are considered however the consistency of the results with previous findings indicate that these findings might apply in other populations." in the discussion section.

Minor:

There are several grammatical errors and errors in punctuation, which require correction.

Abbreviations such as WHO, UCG, need to be explained in full in brackets after the terms.

Answer: Thank you for noting this, we have no carefully proofread the manuscripts and hopefully eradicated the grammatical errors.

Reviewer: 2

Dr. Giulia Cesaroni, ASL Roma 1

Comments to the Author:

The paper "Association between self-reported alcohol consumption and diastolic dysfunction, a cross-sectional study" is a population-based study on alcohol as a risk factor for diastolic dysfunction.

Although the data are from a surveillance of the population in 2002-03, given the high prevalence of both diastolic dysfunction and alcohol consumption, accumulating evidence of an association is important for future prevention.

Here follow some points that, if addressed, could improve the paper.

Major revisions

My major concern about the paper is the statistical approach.

1) The Authors present only the multivariable analyses. It would be useful to see in the same table of multivariable results the results of the models adjusted for age and sex only.

Answer: We agree and in table 2 we have added a model that includes only age and gender.

2) The hypothesis of the role of each variable should be stated and the analysis should be performed accordingly. Which are the confounders? Which the possible effect modifiers? And which are the mediators of the association between alcohol and diastolic dysfunction?

Answer: In the theoretical models we included traditional confounding factors as age and gender. Current smoking status and LTPA are lifestyle factor with strong covariation with alcohol consumption and that can influence the risk for diastolic dysfunction. Traditional determinants of diastolic dysfunction as HT, T2D and obesity were also included in the model. We have incorporated this aspect into the Methods section and explain the rationale behind three models with different set of covariates.

3) If you think that hypertension, diabetes and BMI are possible mediators, you should perform a mediation analysis.

Answer- We did not perform mediation analysis, as there may be several mediators in the casual pathways and the relationships between mediators are complex. The methodology of mediation analysis when several mediators involve is not well-developed. Therefore, we did not include mediation analysis. Instead, we presented results for several models by including different sets of adjustments.

Model 1 includes only the traditional confounding factors i.e. age and gender.

Model 2, in addition, includes life style factors i.e. smoking status and leisure time physical activity (LTPA).

Model 3, including variables, i.e hypertension, T2D and BMI. These variables could be considered as mediators or only regarding them as strong determinants for the diastolic dysfunction, to improve model fit.

By presenting all three models the reader will have the possibility to judge the extent to which a change of the effect measure reflects removing a confounding effect and the extent to which the change of the effect measure reflects the degree of possible mediation.

4) If you think that sex is a possible effect modifier, you should test it and, in any case, you should show the results from stratified analyses (if the interaction is not statistically significant you can show the results in the online material).

Answer: The interaction test was proved to be insignificant and thus we agreed on continuing with adjusted analyses for gender to avoid type II error risk that would increase if analyses were to be done separately for men and women.

5. Finally, in the limitation section the authors should address the fact that the dataset is quite dated, and that the information on alcohol consumption is cross-sectional, not related to the past habit (it could be useful to have in the paper the specific question).

Answer: Thank you for that comment agree. In limitation we have added now: "Moreover, the findings are based on a rural population in Sweden and differences in these association might appear when other population are considered however the consistency of the results with previous findings indicate that these findings might apply in other populations. Finally, this was a cross sectional observation and no conclusions in causality can be drawn and the residual confounding is not possible to rule out."

Minor revisions

Abstract. Add p-trend<0.001 in line 39. Since the confidence intervals overlap it is difficult to believe that there is a dose-response relationship without a p-trend.-Done

Page 3. Spell out OGTT- Done

Page 5. Line 12. Spell out UCG- Done

Page 6. Lines 53-54. Specify in the text the year of the ESC guidelines, and why you have chosen that edition of guidelines. Done

Page 7. Lines 21-27. Reference 25 is not appropriate (there is not a formula, but other references), and the formula is not correctly written in a mathematical notation. For example, could 2.7 be 2.7 ? Thank you. The reference is changed and the formula is written correctly now.

Page 8. Line 20. It is not clear what the three groups are. –

To clarify that we changed to "Being allocated in one of groups based on alcohol consumption was not associated with diastolic dysfunction (p=0.500)(Table1)"

Page 9. Line 11. If you comment a gender specific result, you should give the reader the table too.- The interaction test was proved to be insignificant and thus we agreed on continuing with adjusted analyses for gender to avoid type II error risk that would increase if analyses were to be done separately for men and women.

Page 20. Table 1. I suggest to specify all the categories of categorical variables i.e., women/men, hypertension yes/no, etc. Moreover, please specify the measurement unit from E/A ratio to LVM. I imagine that LVH is LVHa

Answer: Done

VERSION 2 – REVIEW

REVIEWER	Wong, Jorge McMaster University
REVIEW RETURNED	30-Jun-2023

GENERAL COMMENTS	Daka and colleagues submit their revised manuscript titled “Association between self-reported alcohol consumption and diastolic dysfunction, a cross-sectional study” for peer review. The patient population consists of 1038 individuals from a rural Swedish community. In a cross-sectional analysis, the association between categorical alcohol consumption and diastolic dysfunction was examined. Diastolic dysfunction was determined from echocardiographic data and alcohol consumption was self-reported. The authors found that compared to those with “no alcohol” consumption, low and medium-high alcohol was associated with an increased odds of diastolic dysfunction. The manuscript is improved, and it has addressed many issues raised, although some minor issues remain:  1. In the abstract, the authors were asked to move the mean age of participants from the Settings, Participants, and Methods Section to the Results Section, and this was not done. Also, the number of women and men is repeated in both the Settings Section and the Results section, please keep patient characteristics data in the Results section only. 2. Abstract: The authors were also asked to remove the p-value comparing diastolic dysfunction between men and women at baseline, as it doesn’t add to the main message of their paper. In their response, the authors claim they did this, but it was not done. If they choose to leave this, the authors should justify why it is important. 3. Results: Thank you for looking at the relationship between alcohol and LA diameter. Please add the data to the manuscript rather than say ‘data not shown’. 4. Conclusions: Again, cross-sectional studies or cohort studies cannot prove causality. I would remove the term from the conclusions. Can say: “Additional prospective studies ... are needed to improve our understanding of the underlying relationships and mechanisms.” 5. The last sentence of the conclusions is awkward (page 15, line 41). I suggest to replace with “Our results might be of importance in the development of public health policies and preventive strategies.” 6. There remain some minor grammatical/punctuation errors (some examples below):  a. Page 4, line 31: there should be a comma between hypertension and moderate. b. Page 4, line 38: missing comma between drinking and the role. c. Page 9, line 8: “Totally 94%...” should be replaced by “A total of 94%...” d. Page 9, line 24: there should be a comma between “alcohol” and “a higher rate”. 7. Figure 1. Please explain all abbreviations in the figure. For the term M/W 500/538, I suggest to spell out men and women, or delete altogether.
---

VERSION 2 – AUTHOR RESPONSE

Answer to reviewer:

1. In the abstract, the authors were asked to move the mean age of participants from the Settings, Participants, and Methods Section to the Results Section, and this was not done. Also, the number of women and men is repeated in both the Settings Section and the Results section, please keep patient characteristics data in the Results section only.

This has been changed now.

2. Abstract: The authors were also asked to remove the p-value comparing diastolic dysfunction between men and women at baseline, as it doesn't add to the main message of their paper. In their response, the authors claim they did this, but it was not done. If they choose to leave this, the authors should justify why it is important.

It has been removed.

3. Results: Thank you for looking at the relationship between alcohol and LA diameter. Please add the data to the manuscript rather than say 'data not shown'.

Answer: In the result section we have added the results as suggested and it is now "Exploring in the same models the association between alcohol consumption and left atrial diameter no significant differences were observed between group with no consumption and groups with low and moderate to high consumption (Age and sex adjustments: low consumption: mean difference(cm) =-0.04 95% CI - 0.11 - 0.02; moderate to high alcohol consumption mean difference(cm) =-0.02 95% CI -0.10 - 0.07; fully adjusted model: low consumption: mean difference(cm) =-0.02 95% CI -0.08 - 0.04; moderate to high alcohol consumption mean difference(cm) =0.0 95% CI -0.07 - 0.08).

4. Conclusions: Again, cross-sectional studies or cohort studies cannot prove causality. I would remove the term from the conclusions. Can say: "Additional prospective studies ... are needed to improve our understanding of the underlying relationships and mechanisms."

5. The last sentence of the conclusions is awkward (page 15, line 41). I suggest to replace with "Our results might be of importance in the development of public health policies and preventive strategies."

Answer to comments 4 and 5: The section conclusion has been revised and it is now:

"Our study revealed a significant association between alcohol consumption and diastolic dysfunction starting already at low levels that was supported by a dose-dependent pattern. Additional prospective

surveys with adequate measurement of alcohol consumption and possible confounders are needed to improve our understanding of the underlying relationships and mechanisms. Our results might be of importance in the development of public health policies and preventive strategies.”

6. There remain some minor grammatical/punctuation errors (some examples below):

- a. Page 4, line 31: there should be a comma between hypertension and moderate.
- b. Page 4, line 38: missing comma between drinking and the role.
- c. Page 9, line 8: “Totally 94%...” should be replaced by “A total of 94%...”
- d. Page 9, line 24: there should be a comma between “alcohol” and “a higher rate”.

Thank you. This has been changed accordingly.

7. Figure 1. Please explain all abbreviations in the figure. For the term M/W 500/538, I suggest to spell out men and women, or delete altogether.

Answer: Figure 1 has been adjusted accordingly.

8. We note that you have not included some of your responses to reviewer 2 in the manuscript itself. For example, your response to reviewer 2's second comment has not been included in the manuscript. Please revisit reviewer 2's comments and ensure that all of your responses to reviewer 2 are included in the manuscript.

Answer: This is added to the manuscript in section methods subsection statistical methods and it is now:

”In the theoretical models we included traditional confounding factors as age and gender. Current smoking status and LTPA are lifestyle factor with strong covariation with alcohol consumption and that can influence the risk for diastolic dysfunction. Traditional determinants of diastolic dysfunction as HT, T2D and obesity were also included in the model”

VERSION 3 – REVIEW

REVIEWER	Wong, Jorge McMaster University
REVIEW RETURNED	25-Sep-2023
GENERAL COMMENTS	Daka and colleagues submit their revised manuscript titled “Association between self-reported alcohol consumption and diastolic dysfunction, a cross-sectional study” for peer review. The patient population consists of 1038 individuals from a rural

	Swedish community. In a cross-sectional analysis, the association between categorical alcohol consumption and diastolic dysfunction was examined. Diastolic dysfunction was determined from echocardiographic data and alcohol consumption was self-reported. The authors found that compared to those with “no alcohol” consumption, low and medium-high alcohol was associated with an increased odds of diastolic dysfunction. All the issues raised have been satisfactorily answered.
--	---